# Most Commonly-Consumed Food Items by Food Group, and by Province, in China: Implications for Diet Quality Monitoring

**DOI:** 10.3390/nu14091754

**Published:** 2022-04-22

**Authors:** Sheng Ma, Anna W. Herforth, Chris Vogliano, Zhiyong Zou

**Affiliations:** 1National Health Commission Key Laboratory of Reproductive Health, Institute of Child and Adolescent Health, School of Public Health, Peking University, Beijing 100191, China; 2111210073@bjmu.edu.cn; 2Department of Global Health and Population, Harvard T.H. Chan School of Public Health, Boston, MA 02115, USA; anna@annaherforth.net; 3USAID Advancing Nutrition, Washington, DC 20532, USA; chris_vogliano@jsi.com

**Keywords:** diet quality questionnaire, food groups, sentinel foods, dietary diversity

## Abstract

Dietary quality is of great significance to human health at all country income levels. However, low-cost and simple methods for population-level assessment and monitoring of diet quality are scarce. Within these contexts, our study aimed to identify the sentinel foods nationally and by province of 29 food groups to adapt the diet quality questionnaire (DQQ) for China, and validate the effectiveness of the DQQ using data from the China Health and Nutrition Survey (CHNS). The DQQ is a rapid dietary assessment tool with qualitative and quantitative analysis to determine appropriate sentinel foods to represent each of 29 food groups. Dietary data of 13,076 participants aged 15 years or older were obtained from wave 2011 of CHNS, and each food and non-alcoholic beverage was grouped into 29 food groups of the DQQ. The data were analyzed to determine the most commonly consumed food items in each food group, nationally and in each province. Key informant interviews of 25 individuals familiar with diets in diverse provinces were also conducted to identify food items that may be more common in specific provinces. China’s DQQ was finalized based on identification of sentinel foods from the key informant interviews, and initial national results of the quantitative data. Consumption of sentinel foods accounted for over 95% of people who consumed any food item in each food group, at national levels and in all provinces for almost all food groups, indicating the reliability of the sentinel food approach. Food-group consumption data can be obtained through DQQ to analyze dietary diversity as well as compliance with WHO global dietary guidance on healthy diets, providing a low-burden, food-group-based and simple method for China to evaluate diet quality at the whole population level.

## 1. Introduction

Diet quality, including food quality and diversity, is as essential to human health as air is to human life. Poor diet quality is the main contributor to morbidity and mortality worldwide, with micronutrient deficiencies persisting and diet-related non-communicable diseases rising across the world, especially in low-income countries [1,2]. Dietary diversity, one of the best ways for people to meet the requirements for essential nutrients, has long been recognized as a key dimension of high-quality diets [3]. Furthermore, healthy diets must protect against diet-related non-communicable diseases (NCDs) [4]. Appeals to improve diet quality and nutrition call for more programmatic action to measure diet quality, promoting better nutritional status for people.

Many kinds of dietary quality indicators exist as important tools to evaluate the quality of the diet [5]. One major category is nutrient-based indicators, which require quantitative intake of food information and food-nutrient conversion tables to convert food weight into nutrient content, and further compare the nutrient adequacy rate, such as the Healthy Eating Index, the Healthy Diet Indicator, the Diet Quality Index, and the Chinese Healthy Food Diversity (HFD) Index [6,7]. These indicators usually require substantial financial and technical support and are often designed for use in high-income countries/regions with life-cycle assessment databases, which are not available in low and middle-income countries/regions [5,8]. There is a need to be able to quickly and inexpensively monitor diet quality in populations worldwide. Therefore, another category is food/food group-based indicators, one of which that is widely used is Minimum Dietary Diversity for Women (MDD-W), for which only the consumption (not the quantity consumed) of a food group was needed [3]. However, the MDD-W only assesses 10 healthy food groups and has a weak correlation with non-communicable diseases (NCDs), which has brought many restrictions to its own application [4]. Dietary diversity scores, which calculate the number of food groups consumed during a given period, are validated indicators of micronutrient adequacy [9]. Currently, two standard dichotomous indicators correlated to achieving nutrient adequacy are used internationally: one is the minimum dietary diversity (MDD) (≥5 of 8 food groups) [10], while the other is MDD-W (≥5 of 10 food groups) for women of reproductive age (15–49 years) [3,11]. A second indicator, the Global Dietary Recommendations (GDR) score, has been developed as an indicator of diet patterns that protect against NCDs [4]. Based on WHO global recommendations [4], the GDR score consists of two subscales. The first is the GDR-Healthy score, including foods for healthy diets such as whole grains, legumes, dark green leafy vegetables, vitamin A-rich fruits, and nuts and seeds, while the other is the GDR-Limit score, including foods that should be limited to intake due to the high content of free sugar, salt, and total or saturated fat, such as grain-based sweets, deep-fried foods and packaged ultra-processed [12] salty snacks.

These indicators can be useful for helping to formulate intervention strategies and monitor intervention effects [3,13]. However, no consistent, reliable method for gathering food group consumption data had been standardized or adapted for individual countries with a wide variety of diets, including China. There is a need for a valid tool for food group consumption to reflect diet quality in China, promote the establishment of targets, and advocate and formulate policies aimed at improving diets and nutrition.

In the present study, we adapted the diet quality questionnaire (DQQ) for China and used data from the China Health and Nutrition Survey (CHNS) 2011 and qualitative interviews to determine the sentinel food items appropriate for China in each of 29 detailed food groups. Furthermore, we tested the validity of the questionnaire for each province.

The DQQ, a tool for collecting food group level data to reflect healthy dietary patterns at the whole population level [14], uses sentinel foods to capture food group consumption: since it is difficult to enumerate every single food currently available in each group in China, selecting foods known to be major contributors would be suitable. In addition, a previous study has already proven the reliability of the sentinel food method, reporting that using sentinel food data obtained similar results as those using all foods in dietary data collection [4]. This study is not only a part of international work on the validation of diet quality indicators that could be used globally, but also provides a new and more suitable method for China to investigate dietary diversity, helping to shape both national and regional programs and policies effectively.

## 2. Materials and Methods

### 2.1. Quantitative Dietary Data Source

The quantitative data in this study were obtained from wave 2011 of CHNS. The CHNS met the standards for the ethical treatment of participants and was approved by the Institutional Review Committee of the University of North Carolina at Chapel Hill, and the National Institute for Nutrition and Health, Chinese Center for Disease Control and Prevention (2015017). Written informed consent was obtained from all subjects before they participate the survey. Further details of the CHNS data have been previously published [15,16].

The 2011 wave of CHNS included nine provinces (Liaoning, Heilongjiang, Jiangsu, Shandong, Henan, Guangxi, and Guizhou) and three municipalities (Beijing, Shanghai, and Chongqing) that differ in economic development, geography, health indicators, and public resources [17]. Basic data cleaning was performed by the CHNS team before the datasets were uploaded online [18]. Using a multistage random cluster sampling method, 5923 families in 289 communities with over 15,000 participants in total were involved from urban and rural areas. A total of 648 participants with incomplete dietary information were excluded. Our study focused on respondents aged ≥15 years, resulting in the removal of 1927 individuals. Moreover, 74 respondents with implausible intakes (carbohydrate intake greater than 1500 g/d, calcium intake greater than 3000 mg/d, or sodium intake greater than 30 g/d) were further excluded. Finally, 13,076 subjects were included in the analysis. Specific exclusion and inclusion criteria are shown in Figure 1.

The basic sociodemographic characteristics and food consumption information of households were collected by trained interviewers from local Centers for Disease Control and Prevention through face-to-face interviews. Diet was assessed with a combination of 24-h recalls of individual dietary intake and a food inventory at the household level consisting of foods that they consumed at home or away from home in three consecutive days [19]. More detailed information on the dietary survey has been previously reported [20]. Since China’s DQQ is designed to obtain food group intake data of the study population in the past 24 h, dietary intake data in the first day during the three consecutive days were used in this study to verify its validity.

### 2.2. Qualitative Data Collection

Key informant interviews of 25 people from diverse provinces were carried out by the research team. In each interview, key informants were asked to identify or confirm the food items they perceived to be most commonly consumed in each of the 29 DQQ food groups across China, inclusive of regions and seasons. Key informants ranged in occupation from health professionals, government officials, food industry professionals, consumers, expats, and students. These interviewees were familiar with the diets in their local regions, so the most commonly consumed food items in each food group were identified according to the data on how local interviewees name and refer to their local foods. Key informants were also asked to rank the popularity of each food, which influenced where each food item appears in the final DQQ tool question formulation. Each interview lasted 60–90 min and were conducted until consensus was reached. The China DQQ was drafted on the basis of these interviews, and accuracy was confirmed by using the national-level data analysis of the CHNS data.

### 2.3. Data Analysis

All foods and non-alcoholic beverages consumed by subjects were divided into 29 food groups of the DQQ as follows: (1) staple foods made from grains; (2) whole grains; (3) white root/tubers; (4) legumes; (5) vitamin A-rich orange vegetables; (6) dark green leafy vegetables; (7) other vegetables; (8) vitamin A-rich fruits; (9) citrus; (10) other fruits; (11) grain-based sweets; (12) other sweets; (13) eggs; (14) cheese; (15) yogurt; (16) processed meats; (17) unprocessed red meat (ruminant); (18) unprocessed red meat (nonruminant); (19) poultry; (20) fish and seafood; (21) nuts and seeds; (22) packaged ultra-processed salty snacks; (23) instant noodles; (24) deep fried foods; (25) fluid milk; (26) sweetened tea/coffee/milk drinks; (27) fruit juice; (28) sugar-sweetened beverages (SSBs) (sodas); (29) fast food.

To identify sentinel foods, all foods were ranked in descending order according to the consumption frequency contributions in each group nationally and by province, after which cumulative consumption frequency was calculated. Food items that captured consumption of 95% or more of the surveyed individuals, were identified nationally and by each province. Furthermore, the proportion of the population consuming the food group that was captured by the sentinel foods in each food group was calculated. For example, of all people who consumed any vitamin A-rich fruit, 96.5% were captured by the sentinel foods nationally. Data analyses were performed with the Statistical Package for Social Sciences (SPSS), version 25.0 for Windows.

## 3. Results

### 3.1. General Information

The sociodemographic characteristics and food consumption information of a total of 13,076 participants aged ≥15 years are described in Table 1. Nationally, 52.9% of the respondents were female and more than half (57.8%) were from rural villages. The mean age of all participants was 49.7 years with a standard deviation (SD) of 16.0 years old, while the average body mass index (BMI) was 22.8 ± 6.3 kg/m^2^. The average intake of energy was 1929.1 ± 790.2 kcal nationally, with average intakes of carbohydrates, protein, and fat of 273.6 ± 130.8 g, 69.2 ± 34.9 g, and 66.1 ± 45.9 g, respectively. Characteristics for participants of each province are also shown in Table 1.

### 3.2. The DQQ for China

Sentinel foods identified through key informant interviews were used to draft the China-adapted DQQ. They were then checked against draft data analysis at the national level (but not at the province level). Table 2 shows the finalized DQQ for China.

### 3.3. Sentinel Foods Items and Consumption Proportions for Each Food Group, Nationally and by Province

The sentinel food items (represented by “X”) in 29 food groups nationally and by province are shown in Table 3. The quantity and kind of sentinel foods differed by province, due to the diverse dietary habits or different local crops of distinct provinces. Taking the staple food made from grains (group 1) as an example, the sentinel food items in Liaoning Province were rice, steamed buns and bread, whereas those in Henan Province were noodles, steamed buns and bread. However, in almost every food group, different provinces contained at least one sentinel food item in common. For instance, carrots, pumpkin or butternut squash, and sweet potatoes that are orange inside were three national sentinel foods for vitamin A-rich orange vegetables (group 5), and 12 provinces or municipalities had the same or two of the three.

Moreover, in almost all cases the DQQ selected items represented >95% of people who consumed each food group nationally, and over 90% in each province (Figure 2; detailed data shown in Appendix A). For example, in the legumes group (group 4), 98.1% of people consumed the sentinel foods (bean curd or tofu, bean curd sheet, soybean milk, soybeans, and other dried beans) nationally, while same sentinel foods captured >95% of people consuming any legumes in almost every province; only one province (Shanghai) would have required additional sentinel food (soy meat) in this category (shown in Table 3 and Appendix A). Therefore, although there were differences in most common food items between different provinces, the DQQ is valid for use in any province of China, in addition to use at national level.

Food group 1: staple foods made from grains; 2: whole grains; 3: white root/tubers; 4: legumes; 5: vitamin A-rich orange vegetables; 6: dark green leafy vegetables; 7: other vegetables; 8: vitamin A-rich fruits; 9: citrus; 10: other fruits; 11: grain-based sweets; 12: other sweets; 13: eggs; 14: cheese; 15: yogurt; 16: processed meats; 17: unprocessed red meat (ruminant); 18: unprocessed red meat (non-ruminant); 19: poultry; 20: fish and seafood; 21: nuts and seeds; 22: packaged ultra-processed salty snacks; 23: instant noodles; 24: deep fried foods; 25: fluid milk; 26: sweetened tea/coffee/milk drinks; 27: fruit juice; 28: sugar-sweetened beverages (SSBs) (sodas); 29: fast food.

## 4. Discussion

This present study validated the use of the DQQ in China, based on dietary intake data from CHNS 2011. It is the first study to report commonly-consumed food items in each of 29 food groups among Chinese residents by 12 provinces or municipalities, and to determine whether the DQQ sentinel food items are applicable for each province.

The DQQ for China is a low-burden tool to track trends in diet quality over time at low cost and without high technical expertise requirements or large-invested quantitative dietary intake surveys. If implemented widely, the diet quality monitoring data from the DQQ can give researchers and policymakers guidance to promote higher diet quality and heathier diet patterns in certain populations or in certain areas. Dietary data obtained from the DQQ could also be used to evaluate nutrient adequacy and trends of overnutrition, and may be useful in predicting the risk of diet-related NCDs due to its detailed food groups [21]. For China, a country with a long-standing food culture and a wide variety of food, DQQ would be helpful to measure the diet quality and diversity of Chinese people more accurately and more closely to the actual situation at national level. What’s more, similar to a previous study which proved the reliability of sentinel food method [4], our results showed that sentinel foods the DQQ selected captured over 90% of people who consumed each food group in almost every province. Therefore, even if different dietary patterns exist, China DQQ can be effectively applied to each province to evaluate their trend of diet quality and to compare the differences between different provinces, so that decision makers could formulate intervention plans with a targeted approach.

This study has several strengths. First, to our knowledge, this was the first work that attempted to report commonly-consumed Chinese food items in 29 food groups nationally and by province, which could obtain more detailed information on dietary diversity at the population level. Second, the DQQ for China was validated to be effective and could be a low-burden indicator for diet quality measurement without substantial technical and financial support. Third, the development of DQQ for China was part of international collaborative research on diet quality which allowed for comparison among different population groups in different countries or regions. However, several limitations of this study should be noted. First, data collected by DQQ were effective at the population level and can be used to measure and track the diet quality of the population, but are not available for the dietary diagnosis of individuals. Second, DQQ is not designed to collect quantitative intake data, so quantitative surveys such as the CHNS will still be needed periodically to gather nutrient and energy intake information. Third, the application of diet quality indicators based on data from DQQ needs to be further verified in China. The GDR score has been validated in other countries [4]; work is underway to validate it in China. It is also possible to create a composite score which includes both the GDR score and Food Group Diversity Score (FGDS) to reflect nutrient adequacy and the global dietary recommendations, and further to capture the total diet quality [4]. These new scores require further validation, as they may complement or substitute for other indicators in use. Currently, the MDD-W (and its complement for the total population, the FGDS), is the most widely used diet-quality indicator based on food groups, but it did not perform well in reflecting total diet quality or meeting global dietary recommendations [4]. Previous studies reported that in various demographic groups in China, and in pregnant women in Bangladesh, MDD-W led to a high percentage of misclassification, indicating that its applications to other human groups should be well considered [17,22]. The development and application of DQQ may make improvements toward understanding diet quality at population level. Moreover, although this study used data from people over the age of 15, China’s DQQ is developed for general population, including children and adolescents. The validity of China’s DQQ would be validated in children and adolescents in the near future.

## 5. Conclusions

This study first described the most commonly-consumed food items by 29 food groups nationally and by province and developed a low-burden, food group-based DQQ to evaluate diet quality at the population level. The DQQ could be used to evaluate multiple aspects of diet quality and risk factors for NCDs, and could be developed into a tool of low-burden, food group-based indicators, reflecting the dietary quality of the Chinese population.

## Figures and Tables

**Figure 1 nutrients-14-01754-f001:**
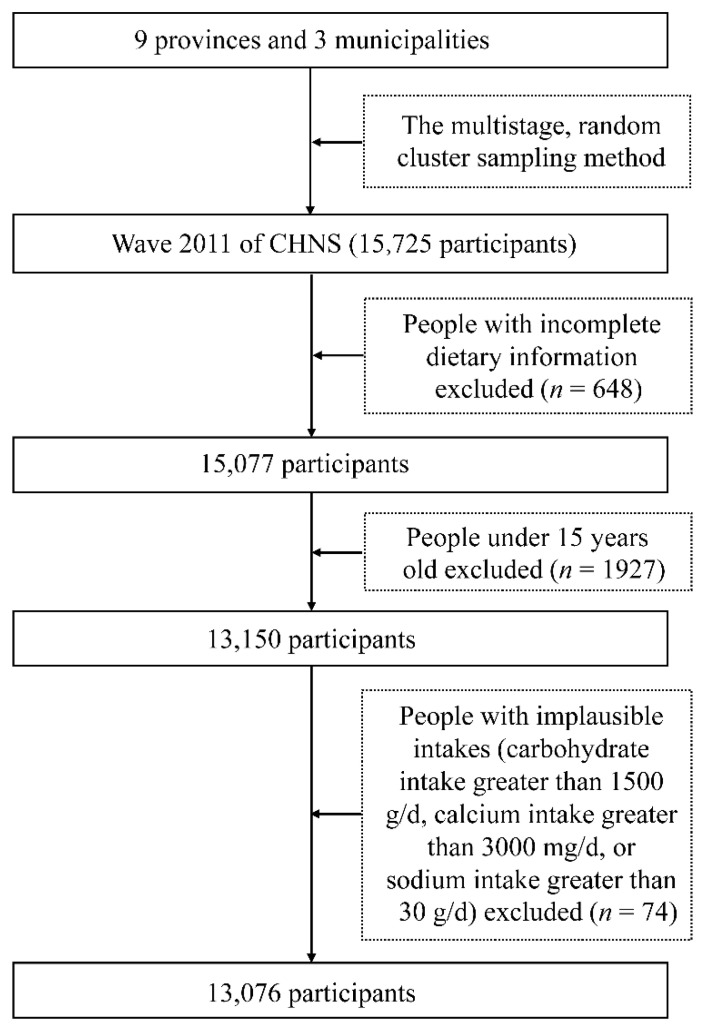
Exclusion and inclusion criteria of this study.

**Figure 2 nutrients-14-01754-f002:**
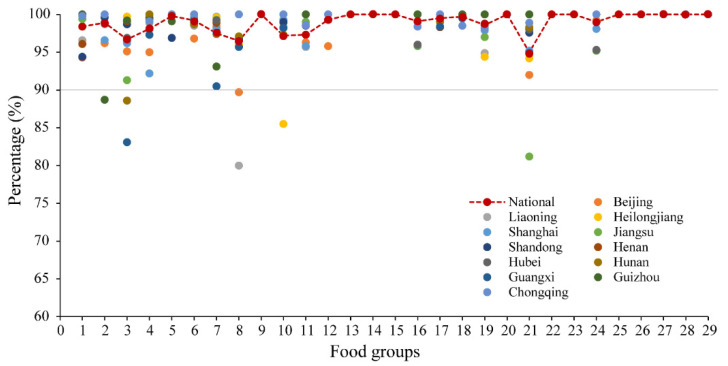
Percent consuming of DQQ sentinel foods in each food group compared to all items, nationally and by province (%).

**Table 1 nutrients-14-01754-t001:** Characteristics of the study population.

Characteristics	National	Beijing	Liaoning	Heilongjiang	Shanghai	Jiangsu	Shandong	Henan	Hubei	Hunan	Guangxi	Guizhou	Chongqing
No. of participants	13,076	1098	978	960	1284	1143	1035	1099	983	1098	1344	1033	1021
Gender, %													
Male	47.1	47.5	47.0	47.7	46.9	46.8	47.0	46.8	46.2	47.0	48.0	47.4	46.2
Female	52.9	52.5	53.0	52.3	53.1	53.2	53.0	53.2	53.8	53.0	52.0	52.6	53.8
Location, %													
Urban	42.2	79.6	28.0	32.0	80.5	33.4	32.4	35.0	32.8	36.1	29.6	30.8	47.7
Rural	57.8	20.4	72.0	68.0	19.5	66.6	67.6	65.0	67.2	63.9	70.4	69.2	52.3
Age, yrs	49.7 ± 16.0	44.8 ± 14.9	52.4 ± 14.7	48.5 ± 14.0	48.7 ± 15.9	51.8 ± 16.7	52.1 ± 15.5	49.0 ± 16.4	51.3 ± 15.0	50.5 ± 15.9	48.1 ± 17.6	51.4 ± 17.1	49.4 ± 14.9
BMI, kg/m^2^	22.8 ± 6.3	24.7 ± 4.6	24.4 ± 5.6	24.1 ± 5.4	24.0 ± 3.9	23.0 ± 5.3	24.9 ± 4.6	20.7 ± 9.2	22.4 ± 6.3	23.1 ± 4.8	18.3 ± 8.9	22.0 ± 5.0	23.3 ± 4.6
Energy, kcal	1929.1 ± 790.2	1661.2 ± 713.2	1926.6 ± 712.7	2071.7 ± 760.8	1771.7 ± 652.6	2160.3 ± 828.5	1941.4 ± 821.9	1833.9 ± 799.3	2265.4 ± 1181.5	2159.8 ± 682.4	1814.1 ± 644.9	1961.0 ± 742.1	1661.6 ± 773.9
Carbohydrate, g	273.6 ± 130.8	231.4 ± 102.6	271.3 ± 112.2	302.4 ± 133.1	212.1 ± 94.8	290.5 ± 132.8	301.3 ± 141.9	309.0 ± 123.6	356.9 ± 196.5	266.8 ± 101.4	301.9 ± 121.9	259.2 ± 99.4	191.0 ± 94.5
Protein, g	69.2 ± 34.9	66.7 ± 35.2	65.6 ± 34.0	66.5 ± 31.5	78.5 ± 35.9	84.0 ± 41.9	70.6 ± 31.6	59.5 ± 26.8	79.6 ± 44.4	72.8 ± 33.3	68.9 ± 27.1	59.7 ± 28.5	54.9 ± 33.2
Fat, g	66.1 ± 45.9	56.6 ± 38.7	67.6 ± 40.1	71.7 ± 37.3	72.0 ± 37.8	77.7 ± 45.6	53.7 ± 36.6	45.8 ± 43.8	61.4 ± 46.1	92.2 ± 47.1	41.7 ± 28.4	79.8 ± 54.0	79.2 ± 60.5

Note: SD, standard deviation; BMI, body mass index.

**Table 2 nutrients-14-01754-t002:** The diet quality questionnaire (DQQ) for China.

**DIET QUALITY QUESTIONNAIRE: CHINA**										
**Read:** Now I’d like to ask you some yes-or-no questions about foods and drinks that you consumed yesterday during the day or night, whether you had it at home or somewhere else. First, I would like you to think about yesterday, from the time you woke up through the night. Think to yourself about the first thing you ate or drank after you woke up in the morning. Think about where you were when you had any food or drink in the middle of the day. Think about where you were when you had any evening meal, and any food or drink you may have had in the evening or late-night, and any other snacks or drinks you may have had between meals throughout the day or night. I am interested in whether you had the food items I will mention even if they were combined with other foods. Please listen to the list of foods and drinks, and if you ate or drank ANY ONE OF THEM, say yes.										
*(Do not read food group names)*		**Yesterday, did you eat any of the following foods?**								*(circle answer)*
*01 Staple foods made from grains*	Global definition	rice	noodles	steamed bun	bread					YES or NO
	Suggested translation	米饭	面条	馒头	面包					
*02 Whole grains*	Global definition	corn	cornmeal	oats	millet	barley	brown rice	black rice	whole wheat bread	
	Suggested translation	玉米(鲜)	玉米面	燕麦片	小米	大麦/糌粑	糙米	黑米	全麦面包	
*03 White roots/tubers*	Global definition	potatoe	lotus root	starch noodles	yam	taro	turnip			YES or NO
	Suggested translation	马铃薯	藕(莲藕)	粉丝/粉条	山药	芋头	大头菜			
*04 Legumes*	Global definition	bean curd or tofu	bean curd sheet	soybean milk	soybeans	other dried beans				YES or NO
	Suggested translation	豆腐	豆腐皮	豆浆	黄豆	其他干豆类				
		**Yesterday, did you eat any of the following vegetables?**								
*05 Vitamin A-rich orange vegetables*	Global definition	carrots	pumpkin or butternut squash	sweet potatoes that are orange inside						YES or NO
	Suggested translation	胡萝卜	南瓜	红薯						
*06.1 Dark green leafy vegetables*	Global definition	Chinese cabbage	water spinach	Chinese spinach	rape	bok choy	sweet potato leaves	broccoli		YES or NO
	Suggested translation	大白菜	空心菜	菠菜	油菜	小白菜	番薯叶	西兰花		
*06.2 Dark green leafy vegetables*	Global definition	mustard leaves	chrysanthemum leaves	radish leaves	amaranth leaves	beet leaves	watercress			YES or NO
	Suggested translation	芥菜	茼蒿	萝卜叶	苋菜	甜菜叶	西洋菜			
*07.1 Other vegetables*	Global definition	cabbage	tomatoes	eggplant	loofah	green bean	local celery	cucumber		YES or NO
	Suggested translation	包菜/圆白菜	番茄	茄子	丝瓜	四季豆	芹菜	黄瓜		
*07.2 Other vegetables*	Global definition	mushrooms	lettuce	radish	cauliflower	seaweed	bamboo shoots	bell pepper	bean sprouts	YES or NO
	Suggested translation	蘑菇	生菜	萝卜	菜花	紫菜	笋	甜椒/柿子椒	豆芽	
		**Yesterday, did you eat any of the following fruits?**								
*08 Vitamin A-rich fruits*	Global definition	persimmon	cantaloupe	ripe mango	passion fruit	fresh or dried apricot	papaya			YES or NO
	Suggested translation	柿子	哈蜜瓜	芒果	百香果	杏或杏干	木瓜			
*09 Citrus*	Global definition	orange	tangerines	pomelo	grapefruit	kumquat				YES or NO
	Suggested translation	橙	橘子/柑橘	柚子	西柚	金桔				
*10.1 Other fruits*	Global definition	apple	pear	watermelon	banana	grapes	kiwi	dragonfruit		YES or NO
	Suggested translation	苹果	梨	西瓜	香蕉	葡萄	猕猴桃	火龙果		
*10.2 Other fruits*	Global definition	jujube	longan	wampee	lychee	pomegranate	cherry	peaches		YES or NO
	Suggested translation	枣子	龙眼	黄皮果	荔枝	石榴	樱桃	桃		
		**Yesterday, did you eat any of the following sweets?**								
*11 Baked sweets*	Global definition	cakes	cookies	sweet pastries	mooncake	rice dumplings	egg tarts			YES or NO
	Suggested translation	蛋糕	甜饼干	甜糕点	月饼	甜粽子/汤圆	蛋挞			
*12 Other sweets*	Global definition	candy	chocolates	jelly pudding	ice cream	popsicles				YES or NO
	Suggested translation	糖果	巧克力	果冻	冰淇淋	棒冰				
		**Yesterday, did you eat any of the following foods of animal origin?**								
*13 Eggs*	Global definition	chicken eggs	preserved duck eggs	quail eggs	pigeon eggs	goose eggs				YES or NO
	Suggested translation	鸡蛋	咸鸭蛋 (白), 松花蛋 (黑)	鹌鹑蛋	鸽子蛋	鹅蛋				
*14 Cheese*	Global definition	cheese								YES or NO
	Suggested translation	奶酪								
*15 Yogurt*	Global definition	yogurt								YES or NO
	Suggested translation	酸奶								
*16 Processed meat*	Global definition	sausages	bacon	ham	larou	luncheon meat	beef jerky	processed beef product	pork jerk	YES or NO
	Suggested translation	腊肠	培根	火腿	腊肉	午餐肉	牛肉干	酱牛肉	猪肉脯	
*17 Unprocessed red meat (ruminant)*	Global definition	beef	lamb sheep or goat	donkey	horse	organs from these animals				YES or NO
	Suggested translation	牛肉	羊肉	驴肉	马肉	内脏 (牛羊驴马)				
*18 Unprocessed red meat (non-ruminant)*	Global definition	pork	pork organs							YES or NO
	Suggested translation	猪肉	猪内脏							
*19 Poultry*	Global definition	chicken	duck	goose	pigeon	chicken gizzard				YES or NO
	Suggested translation	鸡	鸭	鹅	鸽子	鸡胗				
*20 Fish & seafood*	Global definition	fish	seafood							YES or NO
	Suggested translation	鱼	海鲜							
		**Yesterday, did you eat any of the following other foods?**								
*21 Nuts & seeds*	Global definition	sunflower seeds	pumpkin seeds	watermelon seeds	peanut	chestnut	walnuts	almonds	sesame paste	YES or NO
	Suggested translation	葵花子	南瓜子	西瓜子	花生	栗子	核桃	杏仁	芝麻酱	
*22 Ultra-processed packaged salty snacks*	Global definition	chips such as Lays, Pringles	Doritos	shrimp chips	macaroni crisp	spicy strip				YES or NO
	Suggested translation	薯片(乐事, 品客)	立体脆	虾条	通心脆	辣条				
*23 Instant noodles*	Global definition	instant noodles	instant rice noodles							YES or NO
	Suggested translation	方便面	速食米粉							
*24 Deep fried foods*	Global definition	French fries	fried bread stick	fried pancake	fried dough twist	fried glutinous rice ball	fried bean curd	chicken nuggets	deep fried meet	YES or NO
	Suggested translation	薯条	油条	油饼	麻花	炸糕	炸豆腐	鸡块	炸肉	
		**Yesterday, did you have any of the following beverages?**								
*25 Fluid milk*	Global definition	milk	milk powder							YES or NO
	Suggested translation	牛奶	奶粉							
*26 Sweetened tea/coffee/milk drinks*	Global definition	flavored milks	milk tea/bubble tea	nutri-Express	yakult	bottled tea beverage	coffee with sugar			YES or NO
	Suggested translation	果味奶	奶茶/珍珠奶茶	营养快线	酸乳饮料	瓶装茶饮料	咖啡加糖			
*27 Fruit juice*	Global definition	fruit juice	fruit juice beverage							YES or NO
	Suggested translation	果汁	果汁饮料							
*28 SSBs (sodas)*	Global definition	soft drinks such as Coca cola, Pepsi, Fanta, Sprite						sports drink	energy drink	YES or NO
	Suggested translation	软饮料, 如可口可乐, 百事可乐, 芬达, 雪碧						运动饮料	能量饮料	
		**Yesterday, did you get food from any place like?**								
*29 Fast food*	Global definition	KFC	McDonald’s	Pizza Hut	Burger King	Subway	Dicos			YES or NO
	Suggested translation	肯德基	麦当劳	必胜客	汉堡王	赛百味	德克士			

Note: SSB, sugar-sweetened beverages.

**Table 3 nutrients-14-01754-t003:** Sentinel foods for each food group nationally and by province.

**Sentinel Foods**	**National**	**Beijing**	**Liaoning**	**Heilongjiang**	**Shanghai**	**Jiangsu**	**Shandong**	**Henan**	**Hubei**	**Hunan**	**Guangxi**	**Guizhou**	**Chongqing**
**Staple foods made from grains (group 1)**													
rice	X	X	X	X	X	X	X		X	X	X	X	X
noodles	X	X					X	X	X	X·			
steamed buns	X	X	X				X	X					
bread	X	X	X				X	X					
pancake ** in Beijing and Shandong		X					X						
**Whole grain (group 2)**													
millet	X	X	X	X	X	X	X	X	X	X	X		X
cornmeal	X	X	X	X		X	X	X	X		X	X	X
corn	X	X	X	X	X	X		X	X	X	X	X	X
whole wheat bread	X	X	X	X	X					X	X	X	X
oats	X	X			X	X				X			X
barley	X	X			X	X							
black rice	X												X
buckwheat * in Guizhou												X	
**White root/tubers (group 3)**													
potato	X	X	X	X	X	X	X	X	X	X	X	X	X
lotus root	X					X		X	X	X	X		X
starch noodles	X	X			X	X				X		X	X
turnip	X				X	X		X	X	X			X
yam	X	X			X		X			X	X		
taro	X				X	X	X		X	X			X
plantain ** in Jiangsu						X							
plantain * in Guangxi											X		
jicama * in Hunan										X			
**Legumes (group 4)**													
bean curd or tofu	X	X	X	X	X	X	X	X	X	X	X	X	X
bean curd sheet	X		X	X			X	X					X
soybean milk	X	X		X	X	X	X	X	X	X			X
soybeans	X	X			X	X	X	X	X	X	X	X	X
other dried beans	X	X	X	X	X	X	X	X	X	X		X	X
soy meat ** in Shanghai					X								
**Vitamin A-rich orange vegetables (group 5)**													
carrots	X	X	X	X	X	X	X	X	X	X	X	X	X
pumpkin or butternut squash	X	X	X	X		X	X	X	X	X	X	X	X
sweet potatoes that are orange inside	X	X	X		X	X	X	X		X			X
**Dark green leafy vegetables (group 6)**													
Chinese cabbage	X	X	X	X	X	X	X	X	X	X	X	X	X
water spinach	X					X			X		X		X
Chinses spinach	X	X	X	X	X	X	X	X	X	X			
rape	X	X		X	X	X	X	X	X				
bok choy	X	X	X		X	X		X	X	X	X	X	X
sweet potato leaves	X										X		X
broccoli	X	X			X	X							
mustard leaves	X		X							X			
chrysanthemum leaves	X					X							
radish leaves	X								X	X			
amaranth leaves	X									X			X
**Other vegetables (group 7)**													
cabbage	X	X	X	X	X	X	X	X	X	X	X	X	X
tomatoes	X	X	X	X	X	X	X	X	X	X	X	X	X
eggplant	X	X	X	X		X	X	X	X	X	X	X	X
loofah	X	X				X	X	X	X	X	X	X	X
cucumber	X	X	X	X			X	X	X	X	X	X	
local celery	X	X	X	X	X		X	X	X	X	X		
mushrooms	X	X	X	X	X	X				X	X	X	X
lettuce	X	X			X					X	X		X
radish	X		X		X	X	X	X	X	X			X
bean sprouts	X					X	X	X	X		X	X	X
cauliflower	X	X					X	X		X			
bamboo shoot	X				X	X					X		
bell pepper	X	X									X		
seaweed	X									X			
green beans	X	X	X	X				X	X	X	X	X	X
bitter melon ** in Guangxi and Guizhou											X	X	
zucchini ** in Guangxi											X		
chayote ** in Guizhou												X	
**Vitamin A-rich fruits (group 8)**													
persimmon	X	X	X	X	X	X	X	X	X	X		X	X
cantaloupe	X	X		X	X	X			X		X		
ripe mango	X			X							X		
papaya	X				X						X		
hawthorn berry * in Beijing and Liaoning		X	X										
**Citrus (group 9)**													
orange	X	X	X	X	X	X	X	X	X	X	X	X	X
tangerines	X	X	X			X			X	X			
pomelo	X	X	X		X	X		X	X				X
grapefruit	X	X											
kumquat	X							X					
**Other fruits (group 10)**													
apple	X	X	X	X	X	X	X	X	X	X	X	X	X
pear	X	X	X		X	X	X	X	X	X	X	X	X
watermelon	X			X		X					X	X	
banana	X		X	X	X	X	X	X	X	X	X		X
grapes	X	X	X	X		X	X		X	X	X	X	X
kiwi	X				X								X
dragonfruit	X	X			X								
jujube	X		X		X	X	X	X		X			X
longan	X									X	X		
wampee	X										X		
pomegranate	X									X			
lychee	X										X		
cherry	X											X	
peaches	X	X	X	X			X	X			X	X	
honeydew melon * in Heilongjiang				X									
**Grain-based sweets (group 11)**													
cakes	X	X	X		X	X	X	X	X	X	X	X	X
cookies	X	X	X	X	X	X	X	X		X	X	X	X
sweet pastries	X	X	X	X	X	X		X	X	X	X		
mooncake	X		X	X		X	X		X			X	X
rice dumplings	X											X	X
egg tart	X										X		
**Other sweets (group 12)**													
candy	X	X			X	X				X	X		X
chocolates	X	X			X	X		X			X		
popsicles	X	X	X	X		X					X	X	X
ice cream	X		X	X				X			X		
Jelly pudding	X		X		X	X	X	X		X			
**Eggs (group 13)**													
eggs	X	X	X	X	X	X	X	X	X	X	X	X	X
**Cheese (group 14)**													
cheese	X	X			X	X							
**Yogurt (group 15)**													
yogurt	X	X	X	X	X	X	X	X	X	X	X	X	X
**Processed meats (group 16)**													
ham	X	X	X	X	X	X	X	X	X	X	X		
bacon or larou	X				X	X			X	X	X	X	X
sausages	X	X	X	X	X	X	X		X	X	X	X	
beef jerky	X				X								
processed beef product	X	X				X	X						X
pork jerky	X		X		X	X	X	X	X		X	X	
**Unprocessed red meat (ruminant) (group 17)**													
beef	X	X	X	X	X	X	X	X	X	X	X	X	X
lamb sheep or goat	X	X	X		X		X	X		X		X	X
donkey	X			X			X						
organs from these animals	X				X	X			X	X			X
**Unprocessed red meat (non-ruminant) (group 18)**													
pork	X	X	X	X	X	X	X	X	X	X	X	X	X
pork organs	X					X							X
**Poultry (group 19)**													
chicken	X	X	X	X	X	X	X	X	X	X	X	X	X
duck	X	X			X	X	X		X	X	X	X	X
goose	X				X	X							
pigeon	X				X	X							
chicken gizzard	X									X			
chicken heart ** in Liaoning			X										
chicken liver ** in Heilongjiang				X									
**Fish and seafood (group 20)**													
fish	X	X	X	X	X	X	X	X	X	X	X	X	X
seafood	X	X	X	X	X	X	X	X			X	X	X
**Nuts and seeds (group 21)**													
peanuts	X	X	X	X	X	X	X	X	X	X	X	X	X
sunflower seeds	X	X	X	X	X		X	X		X		X	X
chestnuts	X	X	X		X	X	X		X	X		X	X
walnuts	X	X	X		X	X		X		X		X	X
sesame paste	X	X			X					X	X		X
almonds	X	X			X			X		X			
watermelon seeds	X				X		X		X				
pumpkin seeds	X				X							X	
seeds of Euryale ferox ** in Beijing		X											
seeds of Euryale ferox * in Jiangsu						X							
hazelnut ** in Liaoning			X										
pine nut ** in Heilongjiang				X									
pine nut * in Jiangsu						X							
lotus seeds ** in Guangxi											X		
**Packaged ultra-processed salty snacks (group 22)**													
chips	X	X	X		X				X			X	X
shrimp chips	X				X								
**Instant noodles (group 23)**													
instant noodles	X	X	X	X	X	X	X	X	X	X	X	X	X
**Deep fried foods (group 24)**													
fried bread stick	X	X	X	X	X	X	X	X	X	X	X	X	X
fried pancake	X	X	X	X	X	X	X	X	X	X		X	X
fried bean curd	X				X	X				X	X	X	
fried glutinous rice ball	X	X											X
chicken nugget	X							X					X
**Fluid milk (group 25)**													
milk	X	X	X	X	X	X	X	X	X	X	X	X	X
milk powder	X		X	X		X			X	X			
**Sweet tea/coffee/milk drinks (group 26)**													
flavored milk	X	X		X	X	X	X	X		X	X	X	X
Yakult	X	X			X	X	X	X	X	X	X		X
Nutri-Express	X					X					X		
bottled tea beverage	X											X	
**Fruit juice (group 27)**													
fruit juice	X	X	X	X	X	X	X	X	X	X	X	X	X
fruit juice beverage	X	X			X					X			
**SSBs (sodas) (group 28)**													
soft drink	X	X	X	X	X	X		X	X	X	X	X	X
sports drink	X	X			X			X					
energy drink	X									X			
**Fast food (group 29)**													
KFC	X	X	X		X	X	X						
burger	X	X			X		X			X			X
sandwich/sub	X				X	X							
pizza	X	X			X								

Note: “X” represents the sentinel food items of each food group nationally and by province; SSB = sugar-sweetened beverages; * means that the item is not included in the DQQ, and that <90.0% of people were captured by the DQQ sentinel foods in the corresponding province; ** means that the item is not included in the DQQ, and that between 90.0–95.0% of people were already captured by the DQQ sentinel foods in the corresponding province; The DQQ has included luncheon meat in group 16 (processed meat) even though it is not seen in the CHNS, since the people most likely to consume this food are outside of the sample (younger age groups); The DQQ has included horse in group 17 (uprocessed red meat (ruminant)) even though it is not seen in the CHNS, since the people most likely to consume this food are outside of the sample (ethnic minorities living in provinces not included in the survey).

## Data Availability

The dataset in the present study was open-accessed and freely obtained from the CHNS website with registration at https://www.cpc.unc.edu/projects/china/data/datasets/ (accessed on 22 March 2021).

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
