# Peer review of "Most Commonly-Consumed Food Items by Food Group, and by Province, in China: Implications for Diet Quality Monitoring"

_nutrients, 2022, doi:10.3390/nu14091754_

Round 1
Reviewer 1 Report
The study analyzed the quality of the diet through The Diet Quality Questionnaire (DQQ) is a diet assessment tool.
Overall, the Study provided a significant contribution to the field of research. The science underlying the main idea adds significantly to knowledge already published. There are no significant gaps in the cited literature. The presented results are of rilevant importance and impact for the advancement in the field of research. The research design is appropriate.
The findings are important and interesting to the readers. The methods are explained well enough . The discussion section intgrates the findings with relevant theory, rather than simply rehashing the introduction. The writing is good quality.
Author Response
Thank you very much for your positive comments, and we are very encouraged by your recognition of the writing, content and significance of this research.
Reviewer 2 Report
The reviewed manuscript is well written and addresses an important issue related to population dietary analysis.
The manuscript will benefit from a more significant discussion about the results found. For example, lines 187-196 seem more like part of a discussion, which could be expanded in that section.
On the other hand, the tables presented are quite extensive. In that sense, Table 2 could be presented as supplementary material.
Finally, it is necessary to explain the study's objective more precisely, both in the Abstract and in the Introduction.
Author Response
Thank you very much for your comments. We revised the manuscript according to your advice.
The reviewed manuscript is well written and addresses an important issue related to population dietary analysis.
The manuscript will benefit from a more significant discussion about the results found. For example, lines 187-196 seem more like part of a discussion, which could be expanded in that section.
Re: We appreciated the comments. According to your advice, we have expanded the discussion of the results, which can be seen in revised manuscript at lines 242-247.
On the other hand, the tables presented are quite extensive. In that sense, Table 2 could be presented as supplementary material.
Re: Thanks, we agreed that the tables presented are quite extensive, and Table 2 is an important result. As Nutrients is an open-accessed journal online, we would like to keep the Table 2 in the main text, which could be convenient for researchers to have a better understanding of the questionnaire, promoting the application of DQQ for China.
Finally, it is necessary to explain the study's objective more precisely, both in the Abstract and in the Introduction.
Re: According to your suggestion, we revised the Abstract and Introduction sections to explain the study’s objective more precisely. Specific changes have been marked in the resubmitted manuscript, which can be seen at lines 14-17 and lines 78-95.